# A Contemporary Review of Immune Checkpoint Inhibitors in Advanced Clear Cell Renal Cell Carcinoma

**DOI:** 10.3390/vaccines9080919

**Published:** 2021-08-18

**Authors:** Eun-mi Yu, Laura Linville, Matthew Rosenthal, Jeanny B. Aragon-Ching

**Affiliations:** 1GU Medical Oncology, Inova Schar Cancer Institute, Fairfax, VA 22031, USA; eunmi.yu@inova.org; 2Department of Internal Medicine, George Washington University Medical Center, Washington, DC 20037, USA; llinville@mfa.gwu.edu (L.L.); mrosenthal@mfa.gwu.edu (M.R.)

**Keywords:** renal cell cancer, checkpoint inhibitors, immunotherapy, vascular endothelial growth factors

## Abstract

The use of checkpoint inhibitors in advanced and metastatic renal cell carcinomas (RCCs) has rapidly evolved over the past several years. While immune-oncology (IO) drug therapy has been successful at resulting in improved responses and survival, combination therapies with immune checkpoint inhibitors and vascular endothelial growth factor (VEGF) inhibitors have further improved outcomes. This article reviews the landmark trials that have led to the approval of IO therapies, including the Checkmate 214 trial and combination IO/VEGF TKI therapies with Checkmate 9ER, CLEAR, and Keynote-426, and it includes a discussion on promising therapies moving in the future.

## 1. Introduction

Cancers of the kidney and renal pelvis are the sixth most common cancers among men and the ninth most common cancers in women. There will be an estimated number of 76,080 new cases of and 13,780 deaths from these cancers in 2021 [1]. Although there is a wide array of histology in these cancer types, the vast majority of kidney cancers are of clear cell histology [2,3]. The work of the Cancer Genome Atlas Project [4] resulted in the discovery that clear cell renal cell carcinomas (ccRCCs) are defined primarily by mutations in the von-Hippel Lindau/hypoxia-inducible factor (VHL/HIF) pathway, which is directly involved in angiogenesis [3]. Inhibition of the mammalian target of rapamycin (mTOR) pathway is one of several other known mechanisms of carcinogenesis.

Early-stage disease is primarily treated with surgical resection via a partial or radical nephrectomy [5]. Adjuvant therapy with sunitinib can be offered in high-risk cases based on the results of the S-TRAC study [6], though not widely used in clinical practice due to perceived toxicity. While previously there were few effective systemic treatment options available for advanced RCC, we have observed improved outcomes over the past two decades with the development of new anti-VEGF targeted agents and immune checkpoint inhibitors (ICIs). The insights we have gained regarding RCC pathogenesis from the TCGA study have been vital to the development of effective treatment regimens in this disease that has been historically challenging to manage in its advanced stages.

There are several prognostic models that have been proposed for the risk classification of metastatic ccRCC. The most commonly used classification systems in contemporary trials are the Memorial Sloan Kettering Cancer Center (MSKCC) and the International Metastatic RCC Database Consortium (IMDC) models. The MSKCC model established serum lactate dehydrogenase level (LDH), hemoglobin, Karnofsky performance status, corrected serum calcium level, and time from diagnosis to treatment as predictors of outcome based on retrospective data [7]. The IMDC model, initially validated in 2009 and again in 2013, includes the same variables as the MSKCC model, with the exception that neutrophil count and platelet count are used in lieu of serum LDH [8,9]. In both risk models, patients with no negative prognostic factors are considered low or favorable risk. Patients with one or two prognostic factors are placed in the intermediate-risk group, and those with three or more prognostic factors are considered poor risk. Most contemporary interventional trials include subgroup analyses of outcomes based on the prognostic risk group. However, it should be noted that the MSKCC and IMDC models were created and validated prior to the widespread use of ICIs in the treatment of metastatic ccRCC.

In this review, we examine the evolving role of immunotherapy in the treatment of advanced clear cell renal cell carcinoma (ccRCC) and summarize the findings of key clinical trials over the past quarter century. We will conclude with a discussion on what the RCC treatment landscape may look like in the future.

## 2. History of Immunotherapy in RCC

For many years, interferon alfa (IFN-α) and high-dose interleukin-2 (IL-2) were the mainstays of advanced RCC treatment. Treatment options have been limited to cytokine-based regimens because ccRCC is notoriously insensitive to traditional cytotoxic chemotherapy. Though cytokine-based treatments are highly effective in a small subset of patients, response rates are generally low. Responses, if any, are often at the expense of inconvenient drug administration and significant toxicity [10]. Two new classes of drugs were introduced in the systemic treatment of ccRCC within the first decade of the 2000s: mTOR inhibitors and vascular endothelial growth factor receptor (VEGFR) inhibitors. A landmark study compared the use of sunitinib, a small molecule, multi-targeted receptor tyrosine kinase inhibitor, to interferon alfa in the first-line metastatic setting. In this trial, treatment with sunitinib significantly improved progression free survival (PFS) by 6 months, and the objective response rate (ORR) was dramatically improved from 6% with interferon alpha to 31% with sunitinib [11].

In the RECORD-1 trial, everolimus, a novel mTOR inhibitor, was compared with placebo in advanced ccRCC patients who had progressed on prior VEGFR tyrosine kinase inhibitor (TKI) therapy. In the final analysis, treatment with everolimus improved the median PFS by 3 months compared to placebo. The median overall survival (OS) for everolimus and placebo was similar, 14.8 months and 14.4 months, respectively. However, once the survival data were corrected to account for crossover (80% of patients in the placebo arm had crossed over to the everolimus arm), a 1.9-fold increase in overall survival was demonstrated with everolimus [12].

These early clinical trials established the use of VEGF pathway inhibitors as first-line treatment and mTOR inhibitors as second-line treatment for advanced ccRCC.

## 3. Immune Checkpoint Inhibitors as First-Line Therapy

Renal cell carcinoma is one of the most immune-infiltrated tumors, which makes the use of ICIs attractive [13]. Similarly, changes in the microenvironment affect disease biology and responses to systemic therapy [14,15]. The ICIs currently approved for use in the treatment of cancer target the programmed cell death 1 (PD-1) receptor, programmed death ligand 1 (PD-L1), or the cytotoxic T lymphocyte antigen 4 (CTLA4). These drugs have demonstrated efficacy in a variety of malignances, including metastatic melanoma, lung cancers, and Hodgkin lymphoma [16].

The CheckMate 025 trial was one of the first studies to investigate the use of an ICI in advanced ccRCC in the second-line setting. This was a phase 3, randomized, open-label study comparing nivolumab versus everolimus. Patients in this study had already received systemic treatment targeting the VEGF pathway (such as sunitinib or sorafenib). There was a statistically significant difference in OS between the two groups. The patients in the nivolumab group had a median OS of 25 months versus 19.6 months in the everolimus group [17]. The subsequent CheckMate 214 trial investigated dual immune checkpoint inhibition using a PD-1 inhibitor (nivolumab) and a CTLA-4 inhibitor (ipilimumab) compared to sunitinib in the first-line treatment of advanced ccRCC [18]. The primary endpoints in this study were OS, PFS, and ORR. These endpoints all favored the nivolumab plus ipilimumab treatment arm in intermediate- and poor-risk ccRCC patients. ORRs were 42% versus 27% in the combination ICI and sunitinib arms, respectively. At a median follow-up of 25.2 months, the median OS was not reached in the nivolumab plus ipilimumab group, and it was 26 months in the sunitinib group. The median PFS for the nivolumab + ipilimumab arm was 11.6 months versus 8.4 months in the sunitinib arm.

As the use of ICIs gained traction based on the results of the aforementioned ccRCC studies, there was increased interest in combining ICIs with VEGFR TKIs in the treatment of advanced ccRCC (see Table 1 for a list of pivotal first-line trials). The JAVELIN Renal 101 trial studied the use of avelumab, a monoclonal antibody of IgG1 that binds to PD-L1, in combination with axitinib, a small molecule TKI that targets VEGFR, c-KIT, and platelet-derived growth factor receptors (PDGFRs). This combination regimen was compared with sunitinib in the first-line setting for patients with ccRCC, and outcomes were reported in 2019. The median PFS was 13.8 months versus 8.4 months in the avelumab + axitinib and sunitinib arms, respectively. In patients with PD-L1-positive tumors, median PFS was 13.8 months versus 7.2 months in each treatment arm, respectively. The ORR was 55.2% in the avelumab + axitinib arm versus 25.5% in the sunitinib arm [19].

The KEYNOTE-426 trial also compared the combination of an ICI (pembrolizumab) with a VEGFR TKI (axitinib) with sunitinib in patients with advanced ccRCC in the first-line setting. After a median follow-up of 12.8 months, the median PFS was 15.1 months in the axitinib plus pembrolizumab group and 11.1 months in the sunitinib group. The 12-month OS rate was nearly 90% in the combination treatment arm versus 78.3% in the sunitinib arm. There was also a significant improvement in ORR, 59.3% versus 35.7%, in the investigational and sunitinib arms, respectively. Treatment with axitinib plus pembrolizumab was favored in patients across all International Metastatic Renal Cell Carcinoma Database Consortium (IMDC) risk groups. Moreover, the combination regimen proved to be beneficial over sunitinib irrespective of tumor PD-L1 expression [20].

The CLEAR trial compared two different combination regimens with standard-of-care sunitinib monotherapy, the results of which were reported at the 2021 ASCO Genitourinary Cancers Symposium. Patients randomized to the investigational treatment arms received lenvatinib, an oral multi-kinase inhibitor, with either pembrolizumab (L+P) or everolimus (L+E). Each of these treatment arms were evaluated against sunitinib (S) with PFS as the primary endpoint analyzed. A statistically significant improvement in PFS was reported with both combination treatment arms compared to the S arm. The median PFS was 23.9 months in the L+P group, 14.7 months in the L+E group, and 9.2 months in the S group. OS was significantly improved with L+P compared to S (hazard ratio for death, 0.66; 95% CI, 0.53 to 0.80; *p* = 0.005), but there was no statistically significant OS benefit of L+E compared to sunitinib. Objective response rates as determined by an independent review committee for each treatment arm were 71%, 53.5%, and 36.1% in the L+P, L+E, and S arms, respectively [21].

CheckMate 9ER is a phase 3, randomized, open-label trial that evaluated the use of cabozantinib, an oral multi-kinase inhibitor with multiple targets (VEGFR2, c-MET, AXL, and RET), in combination with nivolumab in untreated advanced ccRCC patients. Outcomes were compared to the control group who received standard-of-care sunitinib monotherapy, and the findings from this trial were published in March 2021. At a median follow-up of 18.1 months, the median PFS was 16.6 months for the cabozantinib plus nivolumab treatment arm versus 8.3 months in the sunitinib arm. Combination therapy with cabozantinib plus nivolumab appeared to result in a statistically significant improvement in PFS (hazard ratio for disease progression or death, 0.51, with 95% confidence interval ranging from 0.41 to 0.64; *p* < 0.001), and the 12-month OS probability was 85.7% in the combination treatment arm versus 75.6% in the sunitinib arm. ORR was significantly improved with combination therapy, 55.7% versus 27.1% in those who were treated with sunitinib [22]. Based on these results, cabozantinib in combination with nivolumab was FDA-approved for use in the first-line setting for advanced ccRCC patients in January 2021.

The use of immune checkpoint inhibitors in RCC is generally safe, and while autoimmune toxicities are common, they are manageable when identified early. Milder grade 1 or 2 adverse reactions can be managed by supportive measures (i.e., topical steroid for skin toxicity, thyroid replacement therapy for hypothyroidism), temporary holding of ICI therapy, or low-dose systemic corticosteroid administration. Grade 3 or 4 usually requires higher-dose systemic corticosteroids in addition to holding or permanently discontinuing ICI. Often, toxicity profiles can overlap between ICI and TKI agents. For example, both nivolumab and cabozantinib can cause diarrhea via different mechanisms, making this particular side effect challenging to manage in a patient who is on this combination treatment regimen. Table 2 provides a summary of the toxicity profiles for the first-line immunotherapy regimens that have been discussed.

The real-world treatment of advanced RCC has become complicated given the compelling data reported across all of the aforementioned first-line trials in the past three years. That said, it is a good problem to have several treatment regimens to choose from. It is unlikely we will have data comparing these first-line regimens head-to-head in a single prospective trial to determine which regimen is truly superior; therefore, it is reasonable to offer any of these regimens to an advanced ccRCC patient in the clinical setting. In addition, these different trials set different primary endpoints. Some primarily evaluated progression-free survival as the primary endpoint (for Checkmate 9ER and CLEAR), others used dual primary endpoints such as Keynote 426 or, specifically, the PD-L1 population of patients, in Javelin Renal 101. The choice of therapy, in our opinion, is contingent upon a number of factors including the presence of comorbid conditions (autoimmune disease, cardiovascular disease such as poorly controlled hypertension, pre-existing organ dysfunction) and patient preference as it relates to the treatment schedule and potential toxicities. The management of toxicities can be complicated as there is an overlap of the potential side effects that can occur in TKIs and ICI, for example, diarrhea. When a patient develops diarrhea while on a combination ICI+TKI regimen, it is often difficult to ascertain whether or not it is an immune-mediated toxicity that requires systemic steroids versus simply holding or discontinuing either one or both drugs. Therefore, one could justify a preference for offering nivolumab plus ipilimumab (dual ICI regimen) in the first-line setting based on the CheckMate 214 trial in the appropriate patient as the management of immune-mediated side-effects is more straightforward. On the other hand, adequate resources for education of patients and clinicians, who may not often see such oncology patients who develop autoimmune side-effects but interact with them in various settings (for instance, in the emergency room), is imperative since early recognition of autoimmune toxicity would lead to appropriate treatment and, ultimately, better outcomes. Familiarity with management of autoimmune toxicity as well as drug availability are important considerations as well, along with changes in quality of life parameters that are seen in each of these trials.

## 4. Second-Line Therapy

In spite of the advancements that have been made in the first-line treatment of advanced ccRCC, the majority of patients inevitably require subsequent lines of therapy due to development of disease progression. Several studies have evaluated various systemic single-agent and combination treatment regimens that have changed the landscape of ccRCC management in the second-line setting and beyond. These studies are outlined in Table 3.

The RECORD-1 trial, the results of which were reported in 2008, was the first study that led to the FDA approval of an mTOR inhibitor for second-line use in metastatic ccRCC. In this multi-centered, open-label, randomized controlled trial, prolongation of PFS was observed with those who received everolimus compared to placebo (4.9 months vs. 1.9 months; *p* < 0.0001) after progression during or within 9 months of treatment with a VEGF-targeted therapy (sunitinib, sorafenib, or both). The ORR for everolimus, however, was only 1.8% [23]. The RECORD-3 trial evaluated alternate sequencing of everolimus and sunitinib. At final analysis, reported in 2017, the median combined PFS was 21.7 months with everolimus followed by sunitinib and 22.2 months with sunitinib followed by everolimus. Median OS was 22.4 months for everolimus followed by sunitinib and 29.5 months for sunitinib followed by everolimus [24]. Results from this final analysis support the use of everolimus in the second-line setting after disease progression on first-line sunitinib.

It became clear that mTOR inhibition alone is not highly effective in the treatment of metastatic ccRCC, based on the low response rate reported in the RECORD-1 trial with everolimus monotherapy. mTOR inhibitors have subsequently been studied both compared to and in combination with TKI agents as alternative second-line therapies in ccRCC. In 2015, a significant PFS benefit was reported with the use of everolimus in combination with lenvatinib (a multi-kinase inhibitor) in previously treated advanced ccRCC patients compared to those who received everolimus alone (14.6 vs. 5.5 months; *p* = 0.005) [25]. Similarly, the METEOR trial demonstrated improvements in both PFS and OS with the of use of another multi-kinase inhibitor, cabozantinib, compared to everolimus, which was reported in 2016 [26], leading to the FDA approval of both cabozantinib monotherapy and everolimus in combination with lenvatinib in the second-line setting for advanced ccRCC in 2016. The reported ORR of cabozantinib in the METEOR study was relatively low at 17%. The combination of everolimus and lenvatinib seemed to benefit a greater proportion of patients based on the reported ORR of 43% (22 of 51 patients), but the small sample size was a limitation of this study.

## 5. Immune Checkpoint Inhibitors as Second-Line Therapy

Though mTOR inhibitors and TKIs are active in ccRCC based on the aforementioned studies, there is certainly room for improvement in the second-line space. However, given the increasing use of immune checkpoint inhibition in the first-line setting in recent years, there has not been much activity with regard to ICI use in the second-line setting for advanced ccRCC since the CheckMate 025 trial. As previously mentioned, the CheckMate 025 trial was one of the first trials exploring the use of an ICI in previously treated advanced ccRCC before ICI use became widespread in the first-line setting. In this study, patients with advanced ccRCC who had progressed after one or two lines of antiangiogenic therapy were treated with nivolumab or everolimus. When compared to everolimus, nivolumab demonstrated a significant OS benefit (25 months vs. 19.6 months, *p =* 0.002). Nivolumab also demonstrated superior ORR compared to everolimus (25% vs. 5%; *p* < 0.001) and led to fewer grade 3 or 4 treatment-related adverse events (19% vs. 37%) compared to patients receiving everolimus [17]. Results of this sentinel trial led to use of nivolumab monotherapy as a second-line agent for patients with metastatic ccRCC who have progressed on a TKI in the first-line setting. Recently, a follow-up retrospective analysis from the Turkish Oncology Group Kidney Cancer Consortium (TKCC) database was performed on patients who received nivolumab monotherapy as second-line treatment and beyond. Findings of this study were consistent with the reported results of the CheckMate 025 trial, thus showing that nivolumab improved OS for metastatic ccRCC patients treated in the second-line and beyond [27].

As a result of the positive first-line ccRCC ICI studies, including CheckMate 214, JAVELIN Renal 101, and Keynote 426, there has been an increasing proportion of advanced ccRCC patients previously treated with an ICI by the time second- or third-line therapy is being considered. Conversely, the use of TKI monotherapy is generally limited to those with favorable risk disease or for patients with contraindications to ICI therapy. Given the paucity of data relating to the continuation of immune checkpoint inhibition post-progression on an ICI, patients who have received and progressed on an ICI-based regimen typically move on to receive TKI monotherapy in the second-line setting.

Although limited, there are studies currently re-exploring the use of single-agent ICI, or ICI in combination with other ICIs or TKIs, in patients with ccRCC who have progressed on or after ICI monotherapy, or ICI combination therapy, in the first-line setting. A multicentered, retrospective, cohort study of 69 metastatic RCC patients between 2012 and 2019 assessed the outcomes of rechallenging with an ICI in patients who had previously received an ICI agent. Twenty-nine (42%) patients received first-line ICI in combination with targeted therapy, and 27 (39%) received ICI monotherapy. This study showed an ORR of 23% at ICI rechallenge, compared to an ORR of 37% with ICI exposure in the first-line setting. There were patients who responded to ICI rechallenge regardless of their response to initial ICI therapy, but the likelihood of response to rechallenge was higher among patients who had previously responded to ICI (ORR of 29%) [28]. A similar retrospective analysis published in 2020 looked at the role of salvage ipilimumab and nivolumab in patients with metastatic RCC who had previous exposure to an anti-PD-1 or anti-PD-L1 agent. Ipilimumab and nivolumab combination therapy led to objective responses in a subset of patients with metastatic RCC who had prior exposure to PD-1/PD-L1 inhibitors but were naïve to anti-CTLA-4 antibody [29].

These retrospective data prompted further study of salvage ipilimumab and nivolumab therapy after single-agent nivolumab in subsequent clinical trials. For example, HCRN GU16-260 (NCT03117309) and OMNIVORE (NCT03203473) are evaluating the clinical outcomes of the addition of ipilimumab in patients with metastatic RCC who do not achieve an objective response to nivolumab monotherapy. The phase 2 TITAN-RCC (Tailored immunotherapy approach with nivolumab in advanced renal cell carcinoma) trial studied the use of nivolumab plus ipilimumab as an “immunotherapeutic boost” in 86 intermediate-/poor-risk or metastatic ccRCC patients who experienced progressive disease or stable disease after four doses of nivolumab monotherapy. This study included two independent patient cohorts: those who had no prior therapy (receiving nivolumab first-line) and those previously treated with a TKI agent (receiving nivolumab second-line). The primary endpoint of this study was ORR. Secondary outcomes included remission rate (RR), PFS, and OS. Preliminary results of the study showed that in patients treated in the first-line setting, the ORR to nivolumab alone was 28.7% compared to 37.0% in patients receiving nivolumab followed by the nivolumab/ipilimumab boost. In the second-line setting the ORR was 18.2% among patients receiving nivolumab alone and 28.3% in those receiving nivolumab followed by the nivolumab/ipilimumab boost [30]. The preliminary results of this study support the addition of ipilimumab to nivolumab after initial treatment with nivolumab alone in advanced ccRCC.

In July 2020, an open-label phase III clinical trial, CONTACT-03, began recruitment. CONTACT-03 is studying the combination of an anti-PD-L1 checkpoint inhibitor, atezolizumab, plus cabozantinib versus cabozantinib alone as second- or third-line therapy after prior progression with an immune checkpoint inhibitor. Patients are included if they have a histologically confirmed locally advanced or metastatic ccRCC or nccRCC with radiographic disease progression during or following treatment with a PD-1/PD-L1 inhibitor [31].

PDIGREE (Alliance A031704) is an adaptive phase III trial currently enrolling intermediate-/poor-risk advanced ccRCC patients which will investigate the use of nivolumab and/or cabozantinib after first-line induction with nivolumab and ipilimumab. Treatment of these patients beyond the initial induction phase consisting of four cycles of nivolumab and ipilimumab, as per the CheckMate 214 protocol, will depend on their response to induction therapy. Those who experience a CR with initial induction therapy will continue with nivolumab maintenance. Those patients who have non-CR or non-PD (partial response or stable disease) will receive cabozantinib in addition to nivolumab. Patients who experience progressive disease with induction nivolumab and ipilimumab will subsequently be treated with cabozantinib monotherapy. The primary endpoint for this study will be 3-year OS, and this study will also investigate the use of IL-6 as a potential biomarker [32].

## 6. Future Directions

### 6.1. Immune Checkpoint Inhibition as Adjuvant Therapy

Standard of care treatment for a localized ccRCC includes surgical resection via a partial or radical nephrectomy with curative intent in those who are acceptable candidates for surgery. Most patients with stage 1 and 2 ccRCCs go on an active surveillance protocol after surgery. The rate of recurrent RCC after definitive surgery ranges from 20 to 40% [33]. These high rates may be explained by the presence of micrometastatic disease that is undetectable with current imaging modalities. Therefore, it is worthwhile to consider systemic therapies in the adjuvant setting for localized RCC to determine whether it improves patient outcomes, as is the case with adjuvant therapies for other solid malignancies, such as breast and colon cancers.

Based on the disease-free survival (DFS) benefit demonstrated with 12 months of adjuvant sunitinib in high-risk (pT3, pT4, or node-positive), localized ccRCC patients post-nephrectomy in the S-TRAC trial, sunitinib is an FDA-approved option in the adjuvant setting [6]. An updated analysis of the S-TRAC trial showed a DFS benefit across all patient subgroups, but to date, no OS benefit has been demonstrated [34]. The ASSURE trial investigated the use of adjuvant sunitinib or sorafenib compared to placebo in high-risk ccRCC patients after surgical resection. In contrast to the S-TRAC data, this study did not demonstrate a significant improvement in 5-year DFS [35]. As such, the use of adjuvant sunitinib has not been widely adopted due to the low benefit-to-risk (toxicity) ratio. Adjuvant pazopanib was also studied in a phase III trial of patients with locally advanced RCC at high risk for relapse after nephrectomy, and the results of the primary DFS analysis demonstrated no benefit compared to placebo [36]. To date, no other TKI agents aside from sunitinib have been FDA-approved for use in the adjuvant setting for localized RCC.

At the 2021 ASCO Annual Meeting, promising data regarding the use of adjuvant pembrolizumab from the KEYNOTE-564 were presented. In this trial, patients with intermediate-high risk (pT2, grade 4 or sarcomatoid features, N0 M0; or pT3, any grade, N0 M0), high risk (pT4, any grade, N0 M0; or pT any state, any grade, node positive M0), or M1 NED ccRCC were randomized to receive pembrolizumab 200 mg via intravenous infusion every 3 weeks or placebo for up to 17 cycles [37]. The primary endpoint was DFS, and OS was a secondary endpoint. As of the data cutoff date of 14 December 2020, median follow-up was 24.1 months, and no patients remain on study treatment. At first interim analysis, the primary endpoint of DFS was met, and the estimated DFS rate at 24 months was 77.3% with pembrolizumab versus 68.1% with placebo. The estimated OS rate at 24 months was 96.6% with pembrolizumab versus 93.5% with placebo, but longer-term follow-up is planned for the endpoint of OS [38]. Based on the statistically significant improvement in DFS with adjuvant pembrolizumab demonstrated in this study, it may become the new standard-of-care in this setting.

There are several trials studying the use of other ICIs in the adjuvant setting for localized ccRCC that are underway, including dual ICI regimens (PD-1 or PD-L1 inhibitor in combination with a CTLA-4 inhibitor). Table 4 highlights current ongoing randomized clinical trials exploring checkpoint inhibitors in the adjuvant therapy setting.

### 6.2. Immunotherapy in the Neoadjuvant Setting

Immune checkpoint inhibitors are also being studied in the neoadjuvant setting in RCC patients for whom surgery is planned. There is one phase I clinical trial exploring the safety and efficacy of pre-operative or neoadjuvant nivolumab (NCT02575222), and the phase 3 PROSPER RCC trial (NCT03055013) entails treatment with nivolumab in both the pre- and post-nephrectomy periods. Combination regimens are also being studied in the neoadjuvant setting. The SPARC-1 pilot study (NCT04028245) is currently recruiting localized ccRCC patients to evaluate the combination of canakinumab (ACZ885, IL-1β inhibitor) and spartalizumab (PDR001, PD-1 inhibitor) prior to radical nephrectomy. The phase II Cyto-KIK trial (NCT04322955) is another study evaluating pre-operative combination therapy. In this trial, however, metastatic ccRCC patients for whom a cytoreductive nephrectomy is planned are treated with nivolumab in combination with cabozantinib prior to surgery.

### 6.3. Vaccine Therapy and Other Novel Modalities

Vaccines have also been explored as a new modality for adjuvant RCC therapy. However, to date, no investigational vaccines have demonstrated clear efficacy in RCC. Most recently, a phase III trial assessing the use of an autologous RNA-modulated dendritic cell vaccine AGS-003 (ADAPT, NCT01582672) in combination with adjuvant sunitinib was terminated early due to a clear lack of OS benefit compared to those receiving adjuvant sunitinib alone [39]. There is a currently a phase I/2 trial (NCT00458536) that is studying the safety of a dendritic cell tumor fusion vaccine in combination with granulocyte macrophage colony-stimulating factor vaccine (GM-CSF) in patients with untreated metastatic RCC for whom a cytoreductive nephrectomy is planned. NeoVax, a Personalized NeoAntigen Cancer Vaccine, created with a combination of Neoantigen peptides and Hiltonol (an immunostimulant, Poly ICLC) is being studied alone and in combination with ipilimumab in a phase 1 study of stage III/IV ccRCC patients for whom resection of all known sites of disease is planned prior to receipt of study drug(s).

### 6.4. ccRCC Genomics and Molecular Subtypes

Clear cell renal cell carcinomas represent a very heterogeneous group of tumors. The mechanisms of ccRCC tumorigenesis are complex, but we are gaining a better understanding of molecular or genomic subtypes and the RCC tumor microenvironment (TME) as they relate to treatment response to both TKIs and ICIs. For example, Beuselinck et al. completed a global transcriptome analysis of 53 primary resected ccRCC tumors from patients who developed metastatic disease who were treated with sunitinib in the first-line setting. Four ccRCC molecular subtypes were identified that were associated with differential sensitivity to sunitinib. ccRCC2 and ccRCC3 subtypes were defined by tumors with high pro-angiogenic gene expression and demonstrated increased sensitivity to sunitinib. Conversely, ccRCC1 and ccRCC4 subtypes were defined by c-Myc upregulation and decreased response to sunitinib, with the ccRCC4 subtype being further characterized by an “immune-inflamed” gene signature [40]. Thus, there is likely a greater role for immune checkpoint inhibition in these TKI-resistant RCC subtypes.

The IMmotion-151 phase III study of atezolizumab plus bevacizumab versus sunitinib in untreated metastatic RCC patients demonstrated an improvement in PFS with the combination ICI-based regimen in PD-L1-positive patients. Tumor gene expression analysis was performed by RNA sequencing in 823 patients from the IMmotion-151 trial, and differential outcomes based on T effector/IFNγ and angiogenesis gene expression signatures were demonstrated. A high T effector/IFNγ gene expression signature was associated with PD-L1 expression by IHC, as well as a prolonged PFS in patients treated with atezolizumab plus bevacizumab versus sunitinib. The low angiogenesis gene expressors also appeared to benefit more from atezolizumab plus bevacizumab versus sunitinib [41]. Today, the use of biomarkers and genomic signatures in the real-world management of ccRCC remains investigational but continues to be an area of intense study.

## 7. Conclusions

With the FDA approval of multiple ICI-TKI combination regimens, the treatment landscape for advanced ccRCC continues to transform rapidly. Due to the increasing use of ICIs in the first-line setting, their use in the second and third-line settings is decreasing given the general lack of data to support the use of an ICI beyond progression on one. Although there is an increasing number of TKIs becoming available for use in these settings, further studies are needed to clearly define the use of ICIs after progression on an ICI-based regimen in the first-line setting. The use of ICIs in earlier stages of ccRCC, such as in the pre- and/or post-op settings, shows promise. Furthermore, increasing knowledge of ccRCC tumor heterogeneity, tumorigenesis, resistance mechanisms, and the RCC TME will continue to provide us with new potential therapeutic targets and drug combinations. While we have come a long way from the use of IL-2 to immune checkpoint inhibitors in the treatment of renal cell carcinomas, this is still just the beginning of the immunotherapeutics era in RCC. New breakthroughs in the management of RCC are sure to come through the tireless work of our dedicated scientists, investigators, and patients.

## Figures and Tables

**Table 1 vaccines-09-00919-t001:** Landmark first-line immunotherapy trials in metastatic RCC.

Trial Name	Publication Date	MOA of Investigational Arms	Number of Patients	Primary Endpoint	Results to Date
CheckMate-214	April 2018	PD-1 + CTLA4 inhibitor	1096 (Int/Poor-risk only)	ORR, median OS, and PFS	Nivo+Ipi vs. Sunitinib ORR: 42% vs. 27% Median OS: NR vs. 26 months Median PFS: 11.6 vs. 8.4 months
JAVELIN Renal 101	March 2019	PD-L1 inhibitor and TKI	886	PFS, OS	Ave+Axi vs. Sunitinib Median PFS: 13.8 vs. 8.4 months, 13.8 vs. 7.2 months (PD-L1+) OS: Data pending ORR: 55.2% vs. 25.5% (PD-L1+)
KEYNOTE-426	March 2019	PD-1 inhibitor + TKI	1062	PFS, OS	P+Axi vs. Sunitinib Median PFS: 15.1 vs. 11.1 months 12-month OS: 89.9% vs. 78.3% ORR: 59.3% vs. 35.7%
KEYNOTE-581 (CLEAR)	February 2021	TKI + PD-1 inhibitor or TKI + mTOR inhibitor	1069	PFS	L+P vs. L+E vs. Sunitinib Median PFS: 23.9 vs. 14.7 vs. 9.2 months ORR: 71% vs. 53.5% vs. 36.1%
CheckMate 9ER	March 2021	PD-1 inhibitor + TKI	651	PFS	Nivo+Cabo vs. Sunitinib Median PFS: 16.6 vs. 8.3 months 12-month OS: 85.7% vs. 75.6% ORR: 55.7% vs. 27.1%

RCC = Renal cell carcinoma, PD-1 = programmed cell death protein 1, CTLA4 = cytotoxic T-lymphocyte-associated protein 4, ORR = objective response rate, OS = overall survival, PFS = progression free survival, Nivo-nivolumab, Ipi = ipilimumab, PD-L1 = programmed death-ligand 1, TKI = tyrosine kinase inhibitor, Ave = avelumab, Axi = axitinib, P = pembrolizumab, mTOR = mammalian target of rapamycin, L = lenvatinib, E = everolimus, Cabo = cabozantinib.

**Table 2 vaccines-09-00919-t002:** mRCC first-line immunotherapy trials: toxicity profiles.

	Checkmate 214: N+I		Checkmate 9ER: N+C		CLEAR: L+P		Keynote 426: P+Axitinib		Javelin Renal 101: Avelumab+Axitinib	
All TRAEs, %	All grades	Grade 3–4	All grades	Grade 3–4	All grades	Grade 3–4	All grades	Grade 3–4	All grades	Grade 3–4
Fatigue	37.8%	4.4%	32.2%	3.4%	40.1%	4.3%	38.5%	2.8%	36%	3 (0)
Increased ALT	6.0%	4.0%	28.1%	5.3%	NR	NR	26.8%	13.3%	13%	4 (1)
Hand-foot syndrome	<1%	<1%	40.0%	7.5%	28.7%	4.0%	28.0%	5.1%	33%	6 (0)
Nausea	20.1%	1.5%	26.6%	0.6%	25.8%	2.6%	27.7%	0.9%	25%	1 (0)
Diarrhea	24.0%	4.0%	63.8%	6.9%	61.4%	9.7%	54.3%	9.1%	54%	5 (0)
Decreased appetite	13.9%	1.3%	13.9%	1.3%	40.3%	4.0%	29.6%	2.8%	20%	2 (0)
TRAEs leading to d/c of Rx	22%		19.7% d/c: 6.6% N only; 7.5% C only; 5.6% d/c N+C		37.2% (L: 25.6%; P: 28.7%; 13.4%: both)		D/c of either drug = 30.5%; d/c both drugs = 10.7%; dose reduction of axitinib in 20.3% 4.8%; 18.9%: both)		4.0%	
TRAEs leading to death	1%		<1%		*n* = 15		2.6%		1.0%	

mRCC = metastatic renal cell carcinoma, TRAE = treatment related adverse event, ALT = alanine transaminase, D/C = discontinuation, N = nivolumab, C = cabozantinib, L = lenvatinib, P = pembrolizumab.

**Table 3 vaccines-09-00919-t003:** Landmark second-line mRCC trials.

Trial Name	Date Published	MOA of Investigational Arms	Patient Population	Primary Endpoint	Results to Date
RECORD-1	August 2008	mTOR inhibitor	Pretreated with sunitinib and/or sorafenib	PFS	E vs. Placebo Median PFS: 4.9 vs. 1.9 months
AXIS	December 2011	TKI	Pretreated with sunitinib, bevacizumab plus IFN-gamma, temsirolimus or cytokine	PFS	Axitinib vs. Sorafenib Median PFS: 6.7 vs. 4.7 months
CheckMate 025	November 2015	PD-1 inhibitor	Pretreated with 1–2 regimens of antiangiogenic therapy	OS	Nivo vs. E Median OS: 25 vs. 19.6 months
METEOR	July 2016	TKI	Pretreated with at least one previous VEGFR-TKI	PFS	Cabo vs. E Median OS: 21.4 vs. 16.5 months ORR: 17% vs. 3%
STUDY 205	November 2015	VEGFR-TKI ± mTOR inhibitor	One prior line of VEGFR-TKI	PFS	E+L vs. L vs. E Median PFS: 14.6 vs. 7.4 vs. 5.5 months

mTOR = mammalian target of rapamycin, PFS = progression free survival, E = everolimus, TKI = tyrosine kinase inhibitor, PD-1 = programmed cell death protein 1, OS = overall survival, Nivo = nivolumab, VEGFR = vascular endothelial growth factor receptor, Cabo = cabozantinib, ORR = objective response rate, L = lenvatinib.

**Table 4 vaccines-09-00919-t004:** Ongoing randomized clinical trials for checkpoint inhibitors as adjuvant therapy for localized renal cell carcinoma.

Trial Name	Therapy	Therapy Duration (Years)	Patients	Primary Endpoint	Secondary Endpoints	Anticipated Complete (Year)
IMmotion-010	PD-L1 inhibitor (atezolizumab)	1	Poor-risk RCC following partial or radical nephrectomy	DFS	OS, DFS of patients with 1% PD-L1 expression	2022
Keynote-564	PD-1 inhibitor (pembrolizumab)	1	Int/poor-risk or high-grade clear cell histology without evidence of disease following nephrectomy	DFS	OS; safety and tolerability	2022
PROSPER RCC	PD-1 inhibitor (nivolumab) neoadjuvant and adjuvant	0.75	Localized RCC undergoing nephrectomy (all histology)	EFS (recurrence or death)	OS; safety and tolerability	2023
Checkmate 914	PD-1 ± CTLA-4 inhibitor (nivolumab, ipilimumab)	2	Poor-risk clear-cell RCC following nephrectomy	DFS	OS	2023
RAMPART	PD-L1 ± CTLA-4 inhibitor (durvalumab, tremelimumab)	1	Int/Poor-risk RCC following nephrectomy (all histology)	DFS; OS	MFS; RCC specific survival time	2024

PD-L1 = programmed death-ligand 1, DFS = disease-free survival, OS = overall survival, PD-1 = programmed cell death protein 1, EFS = event-free survival, CTLA4 = cytotoxic T-lymphocyte-associated protein 4, MFS = metastasis-free survival.

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
