# Peer review of "A Contemporary Review of Immune Checkpoint Inhibitors in Advanced Clear Cell Renal Cell Carcinoma"

_vaccines, 2021, doi:10.3390/vaccines9080919_

Round 1

Reviewer 1 Report

Dear Authors,

The review topic is of relevance to the field, however, this review leaves much to be desired and below are a few suggestions :

1) Table 1: please refrain from using qualitative describers of efficacy when numerous quantitative readouts are easily available. Please include a table comparing efficacy readouts across the major trials listed, perhaps, merge the efficacy table with trial design (OS, PFS, ORR published numbers) table 1 to simplify reading.

2) Table 3- same as comment 1 above. Qualitative describers of efficacy endpoints are pretty much useless when comparing across trials that have positive readouts. For all the trials listed, the PFS, OS, ORR are readily available so please include in the table (not just text)

3) When comparing across trials within  sections (for ex. section 3) please include at the end a paragraph that helps the reader understand how each treatment option is or is not being used in clinic i.e. how is the clinical uptake/usage of each treatment option. What are the clinical consideration into using each treatment option and which type of patients is each treatment option most suitable. 

4) Section 6.2 title is in Bold whereas other sub-sections of the same section 6 are not. Please be consistent in formatting.

Good-luck

Author Response

Reviewer 1

Dear Authors,

The review topic is of relevance to the field, however, this review leaves much to be desired and below are a few suggestions :

1) Table 1: please refrain from using qualitative describers of efficacy when numerous quantitative readouts are easily available. Please include a table comparing efficacy readouts across the major trials listed, perhaps, merge the efficacy table with trial design (OS, PFS, ORR published numbers) table 1 to simplify reading.

Response: We thank the Reviewer for this comment.  We have replaced the qualitative description with numerical results of primary endpoints accordingly.

2) Table 3- same as comment 1 above. Qualitative describers of efficacy endpoints are pretty much useless when comparing across trials that have positive readouts. For all the trials listed, the PFS, OS, ORR are readily available so please include in the table (not just text)

Response: We thank the Reviewer for this comment and revised accordingly.

3) When comparing across trials within  sections (for ex. section 3) please include at the end a paragraph that helps the reader understand how each treatment option is or is not being used in clinic i.e. how is the clinical uptake/usage of each treatment option. What are the clinical consideration into using each treatment option and which type of patients is each treatment option most suitable.

Response: We highly appreciate the Reviewer’s comments.  Hence, we have the final paragraph for each section to provide the Authors’ views on the landscape of treatment but we emphasized these are the Authors’ views since there are no (and unlikely ever be) head-to-head comparisons between these different agents, limiting definitive guidance on which agents and treatment option to choose.  Any other statements could be viewed as misleading.

4) Section 6.2 title is in Bold whereas other sub-sections of the same section 6 are not. Please be consistent in formatting.

Response: We thank the Reviewer for this comment, it has already been corrected.

Good-luck

Response: Many thanks

Reviewer 2 Report

This manuscript summarized the current understanding of the immunotherapy strategies of advanced clear cell renal cell carcinoma (ccRCC) using immune checkpoint inhibitors (ICIs) and vascular endothelial growth factor (VEGF) inhibitors (TKI), including clinical trials over the past 25 years leading to approved ICI therapies and ICI/TKI combined therapies. The authors also predicted the future development in RCC immunotherapy. This topic is needed in the area considering ccRCC has poor response to traditional chemotherapy and cytokine-based treatment. This contemporary review is straightforward, and generally well presented in a logical manner. I only have a few minor comments.

  1. Although the authors compared the first-line, second-line immunotherapy RCC trials and their toxicity profiles in Table 1-4, it will be great to rank the treatment regimen in terms of effectiveness and side-effect.
  2. Table 1 is not mentioned in the text.
  3. Table 3 on page 6 line 4: ORR and OS are repeated.

Author Response

Reviewer 2

Comments and Suggestions for Authors

This manuscript summarized the current understanding of the immunotherapy strategies of advanced clear cell renal cell carcinoma (ccRCC) using immune checkpoint inhibitors (ICIs) and vascular endothelial growth factor (VEGF) inhibitors (TKI), including clinical trials over the past 25 years leading to approved ICI therapies and ICI/TKI combined therapies. The authors also predicted the future development in RCC immunotherapy. This topic is needed in the area considering ccRCC has poor response to traditional chemotherapy and cytokine-based treatment. This contemporary review is straightforward, and generally well presented in a logical manner. I only have a few minor comments.

Although the authors compared the first-line, second-line immunotherapy RCC trials and their toxicity profiles in Table 1-4, it will be great to rank the treatment regimen in terms of effectiveness and side-effect.

Response: We highly appreciate the Reviewer’s comment.  However, there would be no simple way to rank these regimens and while there are inherent differences, for instance, better response rates seen in CLEAR (with Lenvatinib and pembrolizumab compared to sunitinib), there was also increased toxicity and the endpoints are different with each trial.  We therefore wrote the caveats in the narrative and emphasized these are the Authors’ opinions, since laying claim for superiority of one regimen over the other would be misleading.

Table 1 is not mentioned in the text.

Response: We thank the Reviewer for this comment.  Reference to Table 1 within the text has been added.

Table 3 on page 6 line 4: ORR and OS are repeated.

Response: We thank the Reviewer for the comment.  Duplicate is removed.

Reviewer 3 Report

This is a very well written review article addressing the clinical efficacy of the immune checkpoint inhibitors in advanced clear cell renal cell carcinoma.

I have some comments:

Renal cell carcinoma is one of the most immune-infiltrated tumors (PMID: 31527133, PMID: 30738745). Emerging evidence suggests that the activation of specific metabolic pathway have a role in regulating angiogenesis and inflammatory signatures (PMID: 32345771, PMID: 28359744). Features of the tumor microenvironment heavily affect disease biology and may affect responses to systemic therapy (PMID: 33265926).

These findings should be referenced and discussed.

Author Response

Reviewer 3

This is a very well written review article addressing the clinical efficacy of the immune checkpoint inhibitors in advanced clear cell renal cell carcinoma.

Response: We thank the Reviewer for this comment.

I have some comments:

Renal cell carcinoma is one of the most immune-infiltrated tumors (PMID: 31527133, PMID: 30738745). Emerging evidence suggests that the activation of specific metabolic pathway have a role in regulating angiogenesis and inflammatory signatures (PMID: 32345771, PMID: 28359744). Features of the tumor microenvironment heavily affect disease biology and may affect responses to systemic therapy (PMID: 33265926).

These findings should be referenced and discussed.

Response: We thank the Reviewer for the comment.  The references have been added accordingly and we allude to RCC as one of the most immune-infiltrated tumors in the text.
